# Effect of Molecular Weight on the Structural and Emulsifying Characteristics of Bovine Bone Protein Hydrolysate

**DOI:** 10.3390/foods12244515

**Published:** 2023-12-18

**Authors:** Yaodi Zhu, Niancheng Hong, Lijun Zhao, Shengnan Liu, Jie Zhang, Miaoyun Li, Yangyang Ma, Dong Liang, Gaiming Zhao

**Affiliations:** 1College of Food Science and Technology, Henan Agricultural University, Zhengzhou 450002, China; zhu_yaodi@163.com (Y.Z.); hongniancheng@163.com (N.H.); zhaolj2014@126.com (L.Z.); 18839004375@163.com (S.L.); zjmj0801@163.com (J.Z.); qiyu17@163.com (Y.M.); liangdong0221@126.com (D.L.); gmzhao@126.com (G.Z.); 2International Joint Laboratory of Meat Processing and Safety in Henan Province, Henan Agricultural University, Zhengzhou 450002, China

**Keywords:** bovine bone protein, molecular weight, structural characteristics, emulsion characteristics

## Abstract

The emulsifying capacity of bovine bone protein extracted using high-pressure hot water (HBBP) has been determined to be good. Nevertheless, given that HBBP is a blend of peptides with a broad range of molecular weights, the distinction in emulsifying capacity between polypeptide components with high and low molecular weights is unclear. Therefore, in this study, HBBP was separated into three molecular weight components of 10–30 kDa (HBBP 1), 5–10 kDa (HBBP 2), and <5 kDa (HBBP 3) via ultrafiltration, and the differences in their structures and emulsifying properties were investigated. The polypeptide with the highest molecular weight displayed the lowest endogenous fluorescence intensity, the least solubility in an aqueous solution, and the highest surface hydrophobicity index. Analysis using laser confocal Raman spectroscopy showed that with an increase in polypeptide molecular weight, the α-helix and β-sheet contents in the secondary structure of the polypeptide molecule increased significantly. Particle size, rheological characteristics, and laser confocal microscopy were used to characterize the emulsion made from peptides of various molecular weights. High-molecular-weight peptides were able to provide a more robust spatial repulsion and thicker interfacial coating in the emulsion, which would make the emulsion more stable. The above results showed that the high-molecular-weight polypeptide in HBBP effectively improved the emulsion stability when forming an emulsion. This study increased the rate at which bovine bone was utilized and provided a theoretical foundation for the use of bovine bone protein as an emulsifier in the food sector.

## 1. Introduction

Oil-in-water emulsions are frequently used in the food industry to make sauces, drinks, and condiments [1]. They can also be used to distribute hydrophobic functional active ingredients like vitamins, minerals, preservatives, and spices [2]. However, an emulsion is a thermodynamically unstable system, and it is normally necessary to choose a suitable emulsifier to help it form a stable emulsion system [3]. The type of emulsifier plays a key role in its emulsifying performance. Currently, synthetic emulsifiers are the main type used in the food industry; however, as a healthy diet becomes increasingly popular, natural emulsifiers are being employed more frequently in food processing [4,5,6].

Bovine bone is the main by-product of livestock slaughter, and it is rich in protein, minerals, fat, and other nutrients. The protein content is as high as 16–25%. Bovine bone protein, as a complete protein, contains eight kinds of amino acids necessary for the human body, and the proportion of amino acids is balanced, which makes it possible to be used as a high-quality natural emulsifier [7,8,9]. However, collagen, the primary component of bovine bone protein, is poorly soluble because of its structural makeup, which restricts the use of bovine bone protein as an emulsifier in the food sector [10,11]. In related studies, the process used to extract proteins was shown to have a major influence on their functional characteristics [12,13]. According to earlier study findings, the bovine bone protein extracted using high-pressure hot water had a superior emulsifying capacity compared to that extracted using conventional acid and alkali. When utilized as an emulsifier to prepare an emulsion, the bovine bone protein extracted using the former approach can form a stable oil-in-water emulsion. Nevertheless, the cause of this phenomenon remains unknown.

Studies have shown that the molecular weight of a peptide or protein plays an important role in the stability of an emulsion because the molecular weight can affect the adsorption behavior of proteins and peptides at the oil–water interface and the properties of the interface layer [14]. For example, the emulsion activity of rice bran protein hydrolysate decreased as hydrolysis increased; that is, a reduction in the molecular weight of rice bran protein resulted in a decrease in its emulsifying capacity [15]. Studies conducted on animal proteins indicated that whey protein peptides with molecular weights > 5 kDa were capable of forming emulsions that were more stable than those with molecular weights < 5 kDa [16]. Additionally, it has been proposed that as the molecular weight of gelatin hydrolysate increases from 18 kDa to 73 kDa, the emulsifying activity index (EAI) of the hydrolysate increases somewhat [17]. On the basis of this, we hypothesized that the bovine bone protein extracted using high-pressure hot water underwent hydrolysis and denaturation at high temperatures and pressures, resulting in a bovine bone protein hydrolysate of different molecular weights that influenced the emulsification properties.

However, it is unclear whether the emulsifying activity of high-molecular-weight and low-molecular-weight components was the same due to the wide range of molecular weight distribution in bovine bone protein extracted using high-pressure hot water (HBBP). Therefore, in this investigation, we employed ultrafiltration to isolate three distinct molecular weight constituents from HBBP based on prior experimental findings: high molecular weight (10–30 kDa), medium molecular weight (5–10 kDa), and low molecular weight (<5 kDa). Subsequently, to elucidate the association between molecular weight and emulsifying capacity, we compared their structural and emulsifying features. This study provided a theoretical basis for the application of bovine bone protein as a natural emulsifier in the food industry to improve its utilization rate.

## 2. Materials and Methods

### 2.1. Materials

Bovine bone was purchased from various biotechnology companies in Hengdu (Zhumadian, Henan, China); 8-anilino-1-naphthalenesulfonate (ANS) was obtained from Sigma-Aldrich (St. Louis, MO, USA); Sodium dodecyl sulfate (SDS), Nile red, and Nile blue were obtained from Suolaibao Biology Science and Technology Co., Ltd. (Beijing, China); soybean oil was purchased from a local supermarket; other reagents were analytically pure.

### 2.2. Extraction of HBBP

Bovine bone was processed into 2–3 cm blocks after fascia and extra fat were removed. Bovine bone was extracted for 4 h at 121 °C and 0.15 MPa following defatting and decalcification, in accordance with a material-to-water ratio of 1:4. The extract was left standing and layered, and after fat separation, protein was obtained after freeze-drying [10].

### 2.3. Fractionation of HBBP

According to the preliminary experimental results, the molecular weight of HBBP was mainly distributed around 30 kDa, which was subsequently separated via ultrafiltration. Ultrafiltration membranes with fluxes of 5 kDa, 10 kDa, and 30 kDa were selected to divide HBBP into three molecular weight components, and the filtrate after each filtration was collected and freeze-dried to obtain the target with the corresponding molecular weight range. Among these, HBBP 1 was designated for 10–30 kDa, HBBP 2 for 5–10 kDa, and HBBP 3 for <5 kDa. The double contraction method was applied to calculate the protein concentration.

### 2.4. Preparation of Emulsions

An aqueous phase was prepared by dispersing 4 g of polypeptide samples into 100 mL of distilled water and stirring for 2 h. The water (75%) and the oil phases (25%) were homogenized for 3 min using a high-speed disperser (T10 basic, IKA, Staufen, Germany), at 10,000 rpm with a S10N-10G homogenizing knife to prepare a crude emulsion. Subsequently, the final emulsion was made using an ultrasonic crusher (JY92-IIN, Xinzhi, Ningbo, China) set to 450 watts for 4 min in an ice-water bath. To inhibit microbial growth, sodium azide (0.01%, *w*/*v*) was added.

### 2.5. Structural Characteristics

#### 2.5.1. Determination of Solubility

A suitable amount of polypeptide sample was weighed and dissolved in distilled water to prepare a solution with a concentration of 3 mg/mL. Of this polypeptide solution, 2 mL was centrifuged at 6500× *g* for 10 min, the supernatant was aspirated as it was from the corresponding stock solution to determine protein concentration, and the solubility was expressed as the percentage of the supernatant protein concentration (mg/mL) to the original protein concentration (mg/mL). The double contraction method was applied to calculate the protein concentration [18].

#### 2.5.2. Determination of Surface Hydrophobicity

The surface hydrophobicity was determined using the ANS fluorescence probe method [19]. A given weight of sample was dissolved in phosphate buffer (pH 7.0) and stirred at room temperature for 1 h with a magnetic stirrer (78-1, Zhongda Instrument Factory, Changzhou, China) through the No. 3 rotor, diluted with the same phosphate buffer to different mass concentrations. A total of 40 μL of 8 mmol/L ANS solution was added to each protein solution (4 mL), and this was shaken and allowed to stand for 3 min. Subsequently, the fluorescence intensity of the sample was determined using a fluorescence spectrophotometer (F-7000, Hitachi, Kyoto, Japan), with an excitation wavelength of 370 nm and an emission wavelength of 490 nm. The initial slope of the protein concentration (mg/mL) vs. fluorescence intensity graph was determined using linear regression analysis, and this served as the surface hydrophobicity index.

#### 2.5.3. Intrinsic Fluorescence Spectrometry

A fluorescence spectrometer (F-7000, Hitachi) was used to measure the intrinsic fluorescence spectrum [20]. The sample was diluted in phosphate buffer (pH 7.0) to generate a solution of 0.1 mg/mL. The emission and excitation wavelengths were set to 290–470 nm and 280 nm, respectively.

#### 2.5.4. Raman Spectra

Raman spectral data were obtained in the range of 1000–4000 cm^−1^ using a Raman spectrometer (LabRAM HR Evolution, Horiba Electronic Technology Co., Ltd., Paris, France). A single crystal of silicon was used to correct the spectral frequency, and the laser was focused on the test sample on the glass slide through a 20× long focusing lens. The spectrum was obtained under the following conditions: three scans, 60-s exposure time, 2 cm^−1^ resolution, 120 cm^−1^/min sampling speed, and data collection once every 1 cm^−1^. The data acquisition for each type of sample was repeated a minimum of three times. The spectrum was smoothed, and the baseline was corrected and normalized using LabSpec version 6.0 [21].

### 2.6. Emulsion Properties

#### 2.6.1. Measurement of the Emulsifying Activity and Stability Indexes

The emulsion activity index (*EAI*) and emulsion stability index (*ESI*) of the samples were determined using spectrophotometry, slightly modified [22]. From the bottom of the freshly made emulsion, 50 μL was taken and diluted 100 times with 0.1% SDS solution, and the absorbance was determined at 500 nm (*A*_0_) using a UV spectrophotometer. The absorbance was measured again after 10 min (*A*_10_). The calculation formulas for the *EAI* and *ESI* were as follows:(1)$$EAI\left(m^{2}/g\right)=\frac{2\times 2.303\times A_{0}\times N}{C\times \varphi \times 10,000}$$
(2)$$ESI\left(\min\right)=\frac{A_{0}}{A_{0}-A_{10}{}_{}}\times 10$$
where *C* is the concentration of the sample (mg/mL), *N* is the dilution multiple (100), *φ* is the volume fraction of the oil phase (0.2), and the numbers 2 and 2.303 are chemical calculation constants.

#### 2.6.2. Measurement of Particle Size Distribution

The droplet size distribution of the emulsions was determined using the Mastersizer instrument (RISE-2008, Runzhi Instruments Ltd., Jinan, China). The detection parameters were as follows: the real part of the refractive index of the medium was 1.33, the dispersion medium was water, the real part of the refractive index of the particles was 1.53, and the imaginary part was 0.10. When the cumulative particle volume accounted for 10%, 50%, and 90% of the total volume, the particle size was represented by D10, D50, and D90, respectively, and Dav represented the average particle size [23].

#### 2.6.3. Confocal Laser Scanning Microscopy

The microstructure of the sample was observed using a confocal laser scanning microscope (A1R HD25, Nikon, Tokyo, Japan) with a 40× objective lens in the fluorescence mode [24]. To dye the oil phase and protein of the emulsion, 50 μL Nile blue (0.01%, *w*/*v*) and Nile red (0.01%, *w*/*v*) solutions were added to 1 mL of emulsion, respectively. After uniform mixing and dark staining for 20 min, Nile Blue and Nile Red images were obtained at excitation wavelengths of 488 nm and 633 nm, respectively.

#### 2.6.4. Percentage of Adsorbed Proteins and Interfacial Protein Concentration

By adapting the procedures of earlier research, the percentage of adsorbed protein (*AP*%) and the interfacial protein (Γ) concentration were determined [25]. In total, 5 mL of sample solution from the bottom of the emulsion was aspirated and transferred into a 10 mL centrifuge tube. The emulsion was centrifuged at 10,000× *g* for 30 min, and the oil and water phases were separated. The clear liquid at the bottom of the sample was collected using a syringe and filtered through a 0.45 μm filter membrane. The protein solution was centrifuged at 10,000× *g* for 30 min, and the protein concentration in the supernatant was determined. The values of the *AP*% and Γ were calculated according to the following formulas:(3)$$AP\%=\frac{\left(C_{s}-C_{f}\right)\times 100}{C_{0}}$$
(4)$$\;\Gamma (\mathrm{mg}/m^{2})=\frac{\left(C_{s}-C_{F}\right)\times d}{6\varphi }$$
where *C*_0_ is the protein concentration in the original emulsion (mg/mL), *CF* is the protein concentration of the water phase after centrifugation (mg/mL), *Cs* is the protein concentration of the emulsion after centrifugation (mg/mL), *d* is the average particle size of the emulsion (μm), and *φ* is the oil phase volume fraction (0.2).

#### 2.6.5. Rheological Properties

The rheological properties of the emulsion were measured using a rotary rheometer (HAAKE Mars60, Thermo Fisher Scientific, Shanghai, China). The apparent viscosity was measured at 25 °C with a steel parallel plate (40 mm diameter, 1 mm gap) in the shear rate range of 0.1–100.0 s^−1^. Subsequently, the dynamic frequency of the emulsion was scanned from 0.01 to 10 Hz, the change in the trends of the storage modulus (G′) and loss modulus (G″) with frequency were recorded, and the change in the surface viscoelasticity was analyzed [26].

### 2.7. Statistical Analysis

The samples were repeated three times, and the measured structure was expressed as the mean ± standard deviation. The data were analyzed applying IBM SPSS 22.0 through Analysis of Variance (ANOVA). The Duncan test and minimum difference were used for back-testing analysis, with *p* < 0.05 considered to represent a significant difference. The data graphs were all created using Origin 2021 software.

## 3. Results and Discussion

### 3.1. Structural Characteristics

#### 3.1.1. Solubility and Surface Hydrophobicity

Protein solubility is a crucial metric for describing the function of proteins, and surface hydrophobicity is a crucial metric for assessing the level of denaturation and structural alterations in proteins [14,27]. As shown in Figure 1a, there were significant differences in the solubility and surface hydrophobicity indexes of peptides with different molecular weight components (*p* < 0.05). With the decrease in molecular weight, polypeptide solubility gradually increased, whereas the surface hydrophobicity of polypeptide decreased with the decrease in molecular weight. This showed that the solubility of polypeptides was negatively correlated with their molecular weight. This was a result of the peptide chain of high-molecular-weight polypeptides having more nonpolar groups, which increases the polypeptide’s hydrophobicity, and the hydrophobic residues exposed on the polypeptide molecule’s surface heavily participating in intermolecular interaction, which decreases the polypeptide’s solubility [20]. Due to the decrease in molecular weight, the peptide bond near the nonpolar group in the peptide chain of low-molecular-weight polypeptides was hydrolyzed, and some hydrophobic regions in its molecular structure were lost, which weakened the hydrophobic interaction, thus reducing the surface hydrophobicity index and increasing the solubility [28].

#### 3.1.2. Intrinsic Fluorescence Analysis

The fluorescence spectrum of a protein or polypeptide mainly reflects the polarity change of the microenvironment where aromatic amino acids are located in its structure [29]. The amplitude of the distinctive fluorescence signal tends to decrease when the environmental polarity around aromatic groups rises, such as when they are exposed to more water [30]. This can be used to indirectly define the change in tertiary structure. Peptides with different molecular weights exhibited clear differences in their maximum absorption peak intensity in their fluorescence spectra (Figure 1b). The peak fluorescence intensity steadily declined in the following order: HBBP 3 > HBBP > HBBP 2 > HBBP 1. This suggested that the aromatic groups in the peptide chains of molecules with various molecular weights were situated in distinct microenvironments. The fluorescence intensity of HBBP 1 was the lowest, which meant that its aromatic groups were exposed to the greatest extent in water. However, the exposure of aromatic groups increased the interaction between peptides and water, which led to an increase in hydrophobicity, which is consistent with the result that HBBP 1 had the highest surface hydrophobicity index [30]. The loss of hydrophobic groups in the HBBP 3 structure reduced the hydrophobic interaction of molecules, which led to a decrease in the polarity of the microenvironment where aromatic groups were located, such that its fluorescence spectrum had the greatest intensity [31].

#### 3.1.3. Raman Spectra

The response signal of the Raman spectrum is mainly realized by reflecting the different vibrations of the amino acid or peptide chain; thus, the Raman spectrum measurement can reflect the secondary structure changes of a protein or polypeptide [32]. Fitting the distinctive amide I band peaks in the Raman spectra of peptides with varying molecular weight components involves using the second derivative and the linear fitting method. To determine the relative presence of secondary structures of peptides with varying molecular weights, each sub-peak was located and examined, and its peak regions as well as the overall peak were computed (Figure 2 and Table 1). The results showed that the polypeptide α-helix, β-folding, and β-rotation contents altered significantly with the change in molecular weight (*p* < 0.05). As the molecular weight of polypeptides decreased, so did the α-helix and β-sheet contents. The decrease in the α-helix content was due to the opening of the tertiary helix structure of the protein under the action of high pressure and heat. The decrease in the β-sheet content was due to the loss of the hydrophobic region of the polypeptide, which reduced the formation of intermolecular hydrogen bonds, with intermolecular hydrogen bonds being the main force maintaining the β-sheet of the polypeptide [33]. These findings demonstrated that polypeptide stability will gradually decline as molecular weight decreases and the structure of the polypeptide changes from order to chaos. Consequently, compared to low-molecular-weight polypeptides, high-molecular-weight polypeptides are more stable.

### 3.2. Properties of Emulsions

#### 3.2.1. Emulsifying Properties

The EAI is a crucial metric for assessing the capacity of proteins or polypeptides to emulsify, which might describe how the protein or polypeptide interacts with fat. In addition, an increase in EAI promotes emulsion formation. The ESI describes the capacity of protein to maintain a stable emulsion; the higher the ESI, the more closely the water and oil phases combine to form an emulsion [34]. As can be seen in Figure 3a, the EAI of the emulsion exhibited a decreasing trend as the polypeptide molecular weight increased, whereas the ESI exhibited an opposite tendency, gradually increasing as the polypeptide molecular weight increased. HBBP 3 had the highest EAI because of its high solubility and more rapid adsorption rate at the oil–water interface, and thus it could better form emulsions. In the comparison of the ESI, that of HBBP 1 was the highest, which was 90.89, while that of HBBP 3 was only 61.41. The reason for this result was that more hydrophobic groups are exposed in the molecular structure of the high-molecular-weight peptides compared with the low-molecular-weight peptides, which promotes the interaction between them and oils, and consequently shows higher emulsion stability.

#### 3.2.2. Particle Sizes in Emulsions

Particle size is one of the most important indices of the stability of reactive emulsions. In the emulsion system, the smaller the droplet size, the better the stability of the emulsion [35]. Figure 3c compares the particle size distribution of polypeptide emulsions with different molecular weights. The results showed that the particle size of each emulsion was in the form of a “single peak”, and the particle size distribution was significantly different (*p* < 0.05), which showed that D10, D50, D90, and Dav decreased significantly with the increase in polypeptide molecular weight (*p* < 0.05, Table 2). Additionally, the volume distribution curve of emulsion particles steadily shifted to the left as polypeptide molecular weight increased (Figure 3b), indicating a general decrease in emulsion particle size. Compared with the average particle size of the HBBP 3 emulsion of 160.09 μm, for the HBBP 1 emulsion, this was only 66.84 μm, which was significantly reduced (Table 2), meaning that HBBP 1 better reduced the aggregation of oil particles. The D10, D50, D90, and Dav of the HBBP 1 emulsion were significantly lower than those of HBBP 3 (*p* < 0.05) (Table 2), suggesting that the total particle size of this emulsion was smaller and its distribution was more uniform. This could enhance the oil droplets’ surface area, raising the likelihood that peptides will adsorb on their surfaces and boosting emulsion stability [36]. Regarding particle size, HBBP 1 had a better emulsifying ability in this study.

#### 3.2.3. Microstructure of the Emulsions

A confocal laser scanning microscope can be used to investigate the distribution of water and oil in different emulsion systems. The microstructure of emulsions prepared with peptides of different molecular weights are shown in Figure 4; oil droplets are green and protein is red. The crimson protein coating around green oil droplets in the four different HBBP emulsions observed in the images suggested that oil/water emulsions were generated by peptides with varying molecular weights. As the molecular weight of the polypeptide increased, the oil droplet particle size of the emulsion system gradually diminished, and the degree of dispersion increased. This demonstrated that the emulsion formed by high-molecular-weight peptides had a more compact and detailed microstructure, which increased the interaction force between the peptides and droplets, and prevented oil droplets from aggregating [37]. This enhanced the stability of the emulsion and was consistent with the findings of earlier studies. This outcome was explained by the fact that, while producing an emulsion, peptides with a higher molecular weight were adsorbed on the surface of oil droplets more tightly than those with a smaller molecular weight, providing enhanced emulsion stability.

#### 3.2.4. Percentage of Adsorbed Proteins and Interfacial Protein Concentration

The higher the adsorption rate of interfacial protein, the less protein or polypeptide was self-absorbed in the system; the higher the adsorption capacity of interfacial protein for the oil; the greater the film thickness formed on the oil droplet surface; and the better the stability of the emulsion formed [14]. The results of protein adsorption at the oil–water interface in the polypeptide emulsions with different molecular weights showed that the interfacial protein concentration and adsorption rate increased progressively as the polypeptide molecular weight rose (Figure 3c). Of these, HBBP 1 had the greatest interfacial protein concentration (27.09 mg/m^2^) and the maximum interfacial protein adsorption rate (28.77%). The rationale for this outcome was that because a high-molecular-weight polypeptide had more hydrophobic groups in its structure, it had a stronger adsorption power (hydrophobic effect) at the oil–water interface [38]. Consequently, this increased the adsorption capacity of HBBP 1 on the surface of oil droplets to form a thicker interface layer, which provided greater spatial repulsion in the emulsion system and improved the stability of the emulsion [39].

#### 3.2.5. Rheological Properties

Rheological characteristics are of great significance to the study of emulsion quality and stability, and the apparent viscosity of general samples is positively correlated with their stability [40]. The viscosity change curve of polypeptide emulsions at a shear rate of 0.1–100 s^−1^ with varying molecular weights is shown in Figure 5a. With the increase in shear rate, the viscosity of the emulsion gradually decreased and finally tended to be stable, showing the phenomenon of shear dilution. With the increase in polypeptide molecular weight, the apparent viscosity of the emulsion gradually increased, and the emulsion prepared with HBBP 1 had the highest apparent viscosity. The reason for this was that high-molecular-weight peptides had a larger adsorption driving force when adsorbed on the surface of oil droplets due to their higher hydrophobic group content [36]. The system’s continuous phase became more viscous as a result of the strong spatial repulsion of the generated emulsion droplets. The higher apparent viscosity weakened the movement speed of fat particles in the system, thus reducing the possibility of droplet aggregation in the system and increasing the stability of the emulsion [36,41].

The dynamic modulus of emulsion can be characterized by the storage modulus G′ and the loss modulus G″. The dynamic frequency scanning findings of various molecular-weight polypeptide emulsions are displayed in Figure 5b. A substantial frequency dependence was demonstrated, whereby G′ for all emulsions was significantly greater than G″ in the shear frequency and increased dramatically with an increase in frequency. The fact that the emulsion’s G′ and G″ did not cross in the frequency range suggested that the emulsion system had produced a weak three-dimensional gel network structure [42]. Because the high-molecular-weight polypeptide contained more hydrophobic groups than HBBP 3, the emulsion formed by HBBP 1 exhibited better elastic deformation when compared to HBBP 3. This was because the increased viscosity of the continuous phase in the system encouraged the formation of a network structure within the emulsion. Such a network structure in an emulsion system improved the deformation resistance and made it more stable [43].

## 4. Conclusions

This work examined the structural variations of bovine bone proteolysis products at various molecular weights and investigated the association between molecular weight and emulsifying capacity. The results showed that the proportion of hydrophobic groups on the polypeptide surface was greater with higher molecular weight (which would increase the overall hydrophobicity), and the α-helix and β-sheet content, representing the ordered secondary structure of protein, was significantly higher than that of the polypeptide with a low molecular weight. This made it possible for high-molecular-weight peptides to provide a stronger adsorption driving force (hydrophobic interaction) when forming emulsions, such that they could form a thicker adsorption layer on the surface of oil droplets. Therefore, high-molecular-weight peptides were significantly superior to low-molecular-weight peptides in reducing particle size and improving viscosity and emulsion stability. This study provided a theoretical basis for the use of bovine bone protein as a natural emulsifier in the food industry. Even though the bovine bone protein hydrolysates in this work were isolated to a certain molecular weight range, they nevertheless contained peptides with a variety of molecular weights. The physical and chemical properties of these peptides were not exactly the same; thus, their actual adsorption characteristics at the oil–water interface need further study.

## Figures and Tables

**Figure 1 foods-12-04515-f001:**
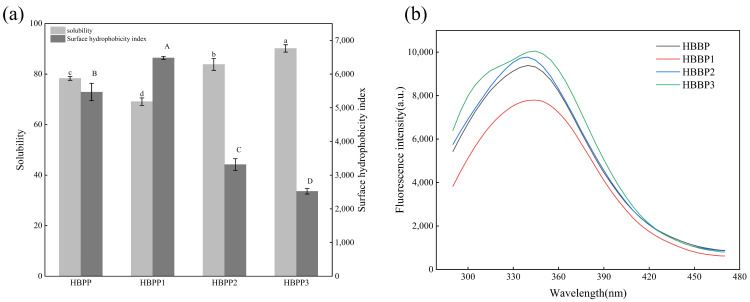
Effect of molecular weight on the solubility, surface hydrophobicity (**a**), and endogenous fluorescence spectrum (**b**) of HBBP. The average values of different letters (a–d, A–D) in the same parameter group were significantly different (*p* < 0.05).

**Figure 2 foods-12-04515-f002:**
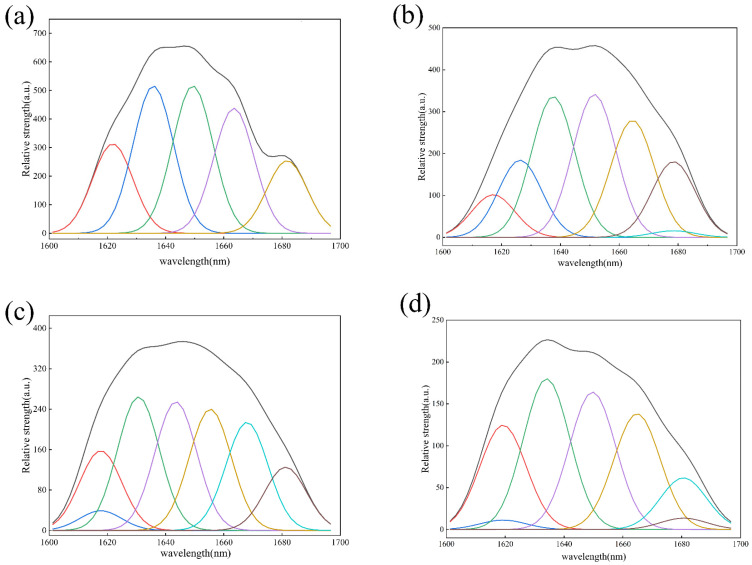
Raman spectrum amide I deconvolution of HBBP (**a**), HBBP 1 (**b**), HBBP 2 (**c**), and HBBP 3 (**d**).

**Figure 3 foods-12-04515-f003:**
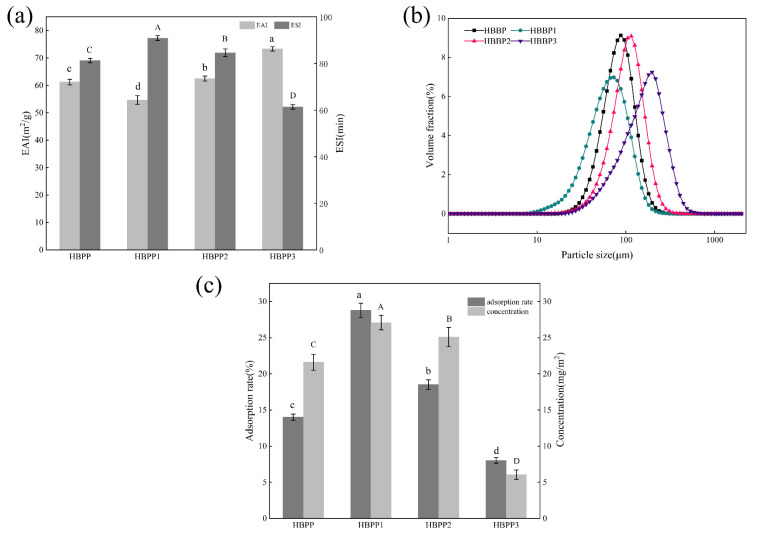
Effect of molecular weight on the emulsifying ability (**a**), particle size distribution (**b**), and interfacial protein adsorption (**c**) of the HBBP emulsions. a–d and A–D indicate that the average values of parameters with different letters in the group were significantly different (*p* < 0.05).

**Figure 4 foods-12-04515-f004:**
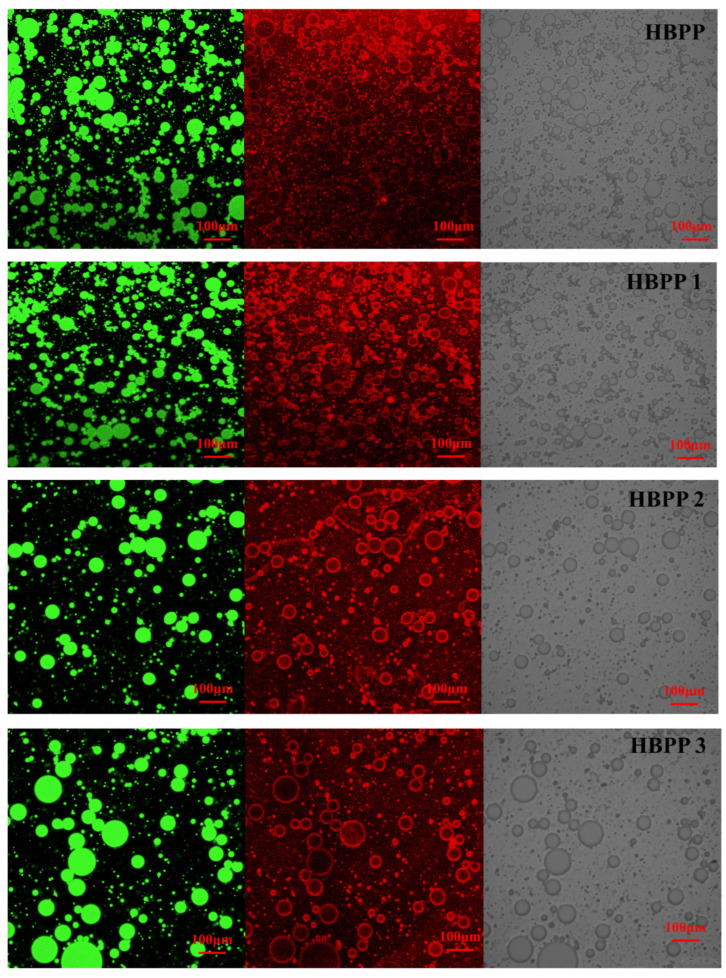
Changes in microstructure images of HBBP emulsions with different molecular weight components using confocal laser scanning, where green represents oil droplets in emulsion, red are protein polypeptide shells, and gray are bright field diagrams.

**Figure 5 foods-12-04515-f005:**
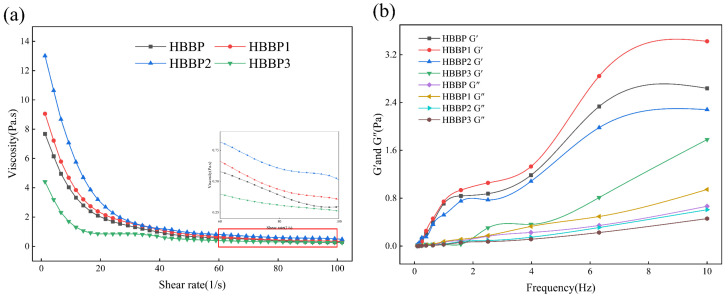
Variation of the viscosity flow curve with shear rate (**a**) and variation of G′ and G″ with angular frequency (**b**).

**Table 1 foods-12-04515-t001:** Secondary structure content of HBBP with different molecular weight components.

Sample	Secondary Structure Content (%)			
	β-Folding	Random Curl	α-Helix	β-Rotation Angle
HBBP	33.74 ± 0.32 ^b^	25.40 ± 0.24 ^c^	21.38 ± 0.34 ^b^	19.49 ± 0.26 ^c^
HBBP 1	35.05 ± 0.24 ^a^	15.36 ± 0.21 ^d^	23.42 ± 0.26 ^a^	25.15 ± 1.88 ^a^
HBBP 2	28.35 ± 0.25 ^c^	26.66 ± 0.31 ^b^	21.51 ± 0.15 ^b^	23.29 ± 0.34 ^b^
HBBP 3	24.54 ± 0.44 ^d^	39.38 ± 0.33 ^a^	18.72 ± 0.27 ^c^	17.28 ± 0.26 ^d^

Note: the results are expressed as means ± standard deviation. For each column, different letters (a–d) indicate that there were significant differences between samples (*p* < 0.05).

**Table 2 foods-12-04515-t002:** Effect of molecular weight on the particle size of the HBBP emulsions.

Sample	Particle Size/μm			
	D10	D50	D90	DAv
HBBP	46.35 ± 0.08 ^c^	80.23 ± 0.03 ^c^	128.05 ± 0.02 ^c^	85.28 ± 0.06 ^c^
HBBP 1	29.18 ± 0.05 ^d^	61.27 ± 0.02 ^d^	111.38 ± 0.11 ^d^	66.84 ± 0.22 ^d^
HBBP 2	56.07 ± 0.03 ^b^	101.75 ± 0.03 ^b^	164.31 ± 0.02 ^b^	107.35 ± 0.03 ^b^
HBBP 3	63.22 ± 0.00 ^a^	149.62 ± 0.01 ^a^	266.58 ± 0.01 ^a^	160.09 ± 0.01 ^a^

Note: the particle sizes of the emulsions are expressed as means ± standard deviation. a–d indicate that there were significant differences in the data in the same group (*p* < 0.05).

## Data Availability

The data used to support the findings of this study can be made available by the corresponding author upon request.

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
