# Peer review of "Effect of Molecular Weight on the Structural and Emulsifying Characteristics of Bovine Bone Protein Hydrolysate"

_foods, 2023, doi:10.3390/foods12244515_

Round 1

Reviewer 1 Report

Comments and Suggestions for Authors

This paper investigates the emulsion properties of bovine bone protein hydrolysate based on differences in molecular weight.

Specify the main protein components of bovine bone protein. Some methods like SDS-PAGE are easier for understanding.

Clarify the meaning of numbers “𝟐×𝟐.𝟑𝟎𝟑” in the EAI formula.

Separate the numerator and denominator in equations (3) and (4) similarly to equations (1) and (2).

Improve the resolution of all figures. Use more large font size.

Figure 4 should add units. And ensure all units are included in figures.

Clearly explain the difference in purpose between the detection using ANS fluorescence probe and intrinsic fluorescence spectrometry in section 3.1.1.

Additionally, elucidate the amino acid composition of HBBP, HBBP1, HBBP2, and HBBP3 as they are examining hydrophobic amino acids. The compositions are important for understanding hydrophobicity.

In sections 2.2 and 2.3, address the possibility of low molecular weight compounds, such as salts and minerals, being present in HBBP3 due to the extraction and fractionation methods. Confirm the composition.

In section 2.5.1, specify the standard used for BCA.

I think BCA is not suitable for low molecular weight compounds, consider using an alternative quantification method.

Please correct the placement of Figures 3b and 3c to their respective correct locations.

Comments on the Quality of English Language

Moderate editing of English language required

spell, space, and  unit missing, etc.

Author Response

Thank you for your review for our manuscript. Please see the attachment.

Reviewer 2 Report

Comments and Suggestions for Authors

Dear authors,

Thank you for the opportunity to read your text. You presented rich and well-documented research material, the results of which I have no objections to. The conclusions are well-documented and confirmed. H

However, I have reservations about the level of English in the text - in many parts, the text must be written from scratch - preferably by a native English speaker.

Moreover, in terms of methodology - you must complete many contradictions in the description of experiments/methodology - so that the reader can repeat your experiments or conduct similar experiments on their own (analogous) systems under precisely the same conditions.

To be completed:

Preparation of emulsions - stirring - please describe the mixer and mixing element homogenization - high-speed disperser (T10 basic, IKA, Staufen, Germany - please describe the type/shape/model of the homogenizing knife

Determination of surface hydrophobicity - mixing - describe the mixer and mixing element.

The emulsion activity index (EAI) and emulsion stability index (ESI) of the samples were determined using previously reported methods[22]. --- this description is not acceptable. The reader could not access your Reference 22 - therefore, you must precisely describe the basics of this research method in the text.

Rheological properties - you presented the "gap" but did not describe the geometry with which the measurements were made: cone-plate / plate-plate / Couette? or some other one?? It must be described - type of geometry and its dimensions - diameters, etc.

Only in the case of rheological measurements did you describe the measurement temperature. For other measurements, you did not present this data.

Comments on the Quality of English Language

English in the text must be improved!

Author Response

(The authors gave the same response as above.)
